# HYBRID NUMERICAL PINNS: ON THE EFFECTIVENESS OF NUMERICAL DIFFERENTIATION FOR NON-ANALYTIC PROBLEMS

## ABSTRACT

This work demonstrates that automatic differentiation has strong limitations when employed to compute physical derivatives in a general physics-informed framework, therefore limiting the range of applications that these methods can address. A hybrid approach is proposed, combining deep learning and traditional numerical solvers such as the finite element method, to address the shortcomings of automatic differentiation. This novel approach enables the exact imposition of Dirichlet boundary conditions in a seamless manner, and more complex, non analytical problems can be solved. Finally, enriched inputs can be used by the model to help convergence. The proposed approach is flexible and can be incorporated into any physics-informed model. Our hybrid gradient computation proposal is also up to two orders of magnitude faster than automatic differentiation, as its numerical cost is independent of the complexity of the trained model. Several numerical applications are provided to illustrate the discussion.

## 1 INTRODUCTION

Solving Partial Differential Equations (PDEs) with deep-learning based approaches has received a growing interest over the past few years, mainly due to the early promising results of Raissi et al. (2019). Their dataless, physics-informed approach, allows to leverage the physical knowledge of a problem to train deep learning models, without having access to numerically costly accurate simulations of the problem.

With this growing interest, many variations of the physics-informed framework have emerged. The main improvements are focused on the model's architecture, the loss construction and the optimization procedure during training. However, few works address the computation of residuals by automatic differentiation, and how some improvements can be made in this direction. In the following, it is argued that this step constitutes a bottleneck of the physics-informed framework and that an hybrid numerical approach could be beneficial, both in terms of global performance and generalization capabilities. More precisely, it is demonstrated in Section 3 that the computation of physical derivatives can lead to wrong results in many settings, for instance when the PDE coefficients do not have an analytic form, or when enriched input data is fed to the deep learning model. In light of these constraints, in Section 4, we propose a new framework, where physical derivatives are computed by numerical operators. A comparison between our new approach and the traditional Physics-Informed framework is presented in Section 5. Finally, we present a numerical experiment in Section 6 to test our hybrid approach for the strong inclusion of Dirichlet boundary conditions on a challenging geometry. We compare our results with the existing Automatic-Differentiation based PINNs, both in terms of accuracy and computation time.

## 2 PRELIMINARIES

**Physics-Informed neural networks** Consider a smooth, open and connex domain $\Omega \in \mathbb{R}^d$, with $d \geq 1$. Consider a general PDE in the following form:

$$\mathcal{N}(u) = f \quad \text{in } \Omega, \tag{1}$$
$$\mathcal{B}(u) = 0 \quad \text{on } \partial\Omega. \tag{2}$$

In this formulation, $\mathcal{N}$ and $\mathcal{B}$ are general, possibly non linear partial differential and boundary operators. The seeked solution is $u$, and $f$ is a given source term.

A neural network with trainable parameters $\theta$ is used to produce a prediction $u_\theta$, which should approximate the true solution $u$. Physics-informed neural networks, as introduced in Raissi et al. (2019) are based on the following framework. In the training phase, the parameters $\theta$ are updated to reach a correct approximation. For this, $N_r$ sample points are sampled inside the domain $\Omega$, and $N_b$ are chosen on the boundary $\partial\Omega$. For a comprehensive discussion on the choice of these points, see for instance Wu et al. (2023). These points are used to approximate the PDE residuals inside the domain, therefore the loss term associated to the partial differential operator $\mathcal{N}$ is:

$$\mathcal{L}_r(\theta) = \frac{1}{N_r} \sum_{i=1}^{N_r} ||\mathcal{N}(u_\theta)(x_i) - f(x_i)||^2. \tag{3}$$

Similarly, the loss term associated to the boundary operator is simply:

$$\mathcal{L}_b(\theta) = \frac{1}{N_b} \sum_{i=1}^{N_b} ||\mathcal{B}(u_\theta)(x_i)||^2. \tag{4}$$

The total loss function $\mathcal{L}(\theta)$ is finally obtained by summing the two loss terms:

$$\mathcal{L}(\theta) = \lambda_r \mathcal{L}_r(\theta) + \lambda_b \mathcal{L}_b(\theta). \tag{5}$$

The weighting coefficients $\lambda_r$ and $\lambda_b$ are hyperparameters allowing to balance the two loss terms during training. They play a key role in the performance of the neural network and are often hard to determine in practise. For a thorough study of these parameters, see Wang et al. (2021a).

Physics-informed models have proven to be very effective in several academic cases (Raissi et al., 2019; Kharazmi et al., 2021), and many variations of these models have been applied to a wide range of problems, such as fluid and solid mechanics (Winchenbach & Thuerey, 2024; Chenaud et al., 2024), climate modelling (Bonev et al., 2023), biology (Yazdani et al., 2020) and many others. For an overview of physics-informed models and their applications, see, for instance, Cuomo et al. (2022); Karniadakis et al. (2021). However, unlike their traditional numerical solver counterparts, such as finite element methods, they lack theoretical guarantees of convergence. Recent works, including Hong et al. (2021); Siegel & Xu (2020), have made some first steps in this direction.

Since the first attempts at solving PDEs with neural networks, which can be traced back to Dissanayake & Phan-Thien (1994); Lagaris et al. (1998a;b), and especially since the work of Raissi et al. (2019), many efforts have been conducted to enhance the physics-informed framework. Leake & Mortari (2020); van der Meer et al. (2022); Berg & Nyström (2018); Wang et al. (2021a); Sheng & Yang (2022) have addressed the complex issue of balancing the different loss terms accounting for the PDE residuals and the data knowledge (boundary or initial conditions for instance) to ensure an efficient training. In a similar fashion, weak PDE formulations have been used too (Samaniego et al., 2020; Zang et al., 2020). Other approaches have investigated more suitable model architectures, using graph networks (Gao et al., 2022; Belbute-Peres et al., 2020; Pfaff et al., 2020) or other machine learning models (Dong & Li, 2021; Geneva & Zabaras, 2020; 2022). For generalization purposes, some works have been focusing on enriching the model, through the loss function (Yu et al., 2022) or by changing the learning task to learn operators instead of functions to a single problem (Wang et al., 2021b; Podina et al., 2023).

Finally, closer to this paper, some works, such as Belbute-Peres et al. (2020); Meethal et al. (2023); Xiang et al. (2022a); Gao et al. (2022) have combined numerical methods with the deep learning approach. However, in Belbute-Peres et al. (2020); Meethal et al. (2023), the idea was not to compute the loss terms differently, but rather to decompose the problem of solving the PDE between the numerical solver and the deep learning model. In Lim et al. (2022); Xiang et al. (2022b;a); Gao et al. (2022), the residuals are computed without automatic differentiation, using respectively classical, hybrid and generalized finite difference methods, and discretized Galerkin analytic formulations. However, to the best of our knowledge, these works did not emphasize the theoretical limitations of automatic differentiation for loss computation. Some works, such as Meethal et al. (2023); Eshaghi et al. (2024) present a hybrid Finite Element- Physics-Informed framework. Howedver, instead of extracting the gradient operator, which we do in this work, they use the whole Finite Element formulation of the PDE as a loss term, which must be built for every instance of a PDE. We argue that our approach is more general, since the operator only depends on the geometry and the mesh, and can be reused for other limit conditions and PDEs.

**Automatic differentiation** (AD) plays now a key role in machine learning and deep learning. There are many ways of implementing this set of techniques, and each deep learning framework, such as Pytorch (Paszke et al., 2019), Tensorflow (Abadi et al., 2015) or JAX (Frostig et al., 2018) has its own specificities. Here, we only describe briefly the *reverse mode* AD, since it is the most common in machine learning settings. For a broader overview of AD and its applications, see Baydin et al. (2018).

AD is based on the fact that computational operations are obtained by combinations of a finite number of elementary, differentiable operations, for which the analytical derivatives are known. In reverse mode AD, computations are decomposed into two phases: a forward and a backward phase. In the forward phase, every computation made is recorded hierarchically, inside a *computational graph*. This register allows to pile up all the dependencies between the variables. In the backward phase, the registered operations are run through backward, from the outputs to the inputs. At each step, the derivative of the associated elementary operation is computed in a formal way, and evaluated at the corresponding value. Next, with the chain rule, the derivatives are multiplied, to obtain as a global result the derivative of the outputs, with respect to every input. This allows for any gradient computation, for instance to compute the derivative of a neural network's output with respect to its parameters. Since most of the optimization processes in deep learning are gradient-based and are variations of the gradient descent procedure, AD has proven to be very efficient and succesful in these applications.

## 3 WHEN AND WHY AUTOMATIC DIFFERENTIATION FAILS

### 3.1 NON-ANALYTICAL PDE COEFFICIENTS

While many applications of physics-informed models have been published recently, the vast majority of the cases are academic, analytical PDEs, where the target solution and the PDE coefficients have a known, analytical expression. However, for many real-life cases in Physics or engineering, the PDE coefficients are not of this type, and can only be tabulated. For instance, this may be the case for thermal and electrical conductivity for diffusion processes, viscosity in fluid mechanics, or material properties in solid mechanics. When this situation arises, the computation of the PDE residuals by AD is compromised. As an example, consider the following one-dimensional static diffusion equation:

$$-(\alpha(x)u')' = 0. \tag{6}$$

Suppose the diffusion coefficient $\alpha$ is given by tabulated data, and is not known analytically. Consequently, the diffusion term of (6) will be computed as $-\alpha(x)u''$, since the spatial derivative of $\alpha$ cannot be computed by the AD framework. Therefore, in this case, the PDE residuals computed by AD will not be accurate, preventing the model from converging to the true solution. This limitation is illustrated in Section 5.

## 3.2 Strong imposition of Dirichlet boundary conditions

Respecting boundary and limit conditions is a challenge for Physics-Informed models. Many works have focused on weak or strong imposition of boundary conditions, first for simple geometries (Lagaris et al., 1998a), then for more complex ones (Leake & Mortari, 2020; van der Meer et al., 2022; Berg & Nyström, 2018; Wang et al., 2021a; Sheng & Yang, 2022; Sukumar & Srivastava, 2022). While these methods can give accurate results, they often require additional preprocessing and computations. Moreover, weak imposition approaches do not guarantee an exact satisfaction of these boundary conditions. On the other hand, classical numerical approaches can usually impose Dirichlet boundary conditions exactly, by modifying the linear system of equations to account for these conditions. A similar approach, within the physics-informed framework, would be to proceed as follows.

Suppose one wants to solve the following PDE, with Dirichlet boundary conditions:

$$\mathcal{N}(u) = f \quad \text{in } \Omega, \tag{7}$$
$$u = u_D \quad \text{on } \partial\Omega. \tag{8}$$

Here, $\Omega$ can be any arbitrary open and connex domain of $\mathbb{R}^d$, $d \geq 1$, and $u_D$ is the Dirichlet boundary condition. Suppose a neural network $M_\theta$ is used to approximate the solution $u$. A simple, straightforward way of strongly imposing the boundary conditions is to simply modify the neural network's outputs when on the boundary. The predicted solution $\hat{u}$ of the PDE (7) - (8) would therefore simply be:

$$\hat{u}(x) = (1 - \mathbb{1}_{\partial\Omega}(x))M_\theta(x) + \mathbb{1}_{\partial\Omega}(x)u_D(x). \tag{9}$$

Here, $\mathbb{1}_{\partial\Omega}$ is the characteristic function of the set $\partial\Omega$.

While this very simple trick guarantees that the boundary conditions are fulfilled, to the best of our knowledge, no work has been published using this approach for Physics-Informed Neural Networks. Once again, the reason comes from the inability of AD to include spatial fields (in this case, $\mathbb{1}_{\partial\Omega}$, and eventually $u_D$ if the Dirichlet conditions do not have an analytical expression) that have not been recorded inside of its computational graph. Therefore, the gradient of $\hat{u}$ will be computed by AD as:

$$\nabla\hat{u} \underset{\text{auto. diff.}}{=} (1 - \mathbb{1}_{\partial\Omega}(x))\nabla M_\theta(x). \tag{10}$$

While this computation is correct on the interior of the domain, this leads to singularities close to the boundary, and the Dirichlet condition (8) will not be taken into account in the training. In particular, for a PDE where the zero function (or any constant function) would be a solution to (7), but not to (8), an AD-based training would converge to such solution. For an example of this behavior, along with a comparison with our proposed framework, which is described in Section 4, see Section 5.2. Conceptually, our approach intends to demonstrate that the model does not need to learn explicitly the boundary conditions. Rather, by strongly applying the BC as external unknowns, the model can expand its capacity to generalization as it should learn how to propagate the information throughout the domain.

Other conditions, such as Neumann or Robin conditions, may not be addressed as directly as what has been proposed here. For these conditions, a variational approach could be considered, in the same fashion as Samaniego et al. (2020). For periodic conditions, a preprocessing step could be implemented. For a more thorough discussion on this point, see Wang et al. (2023).

## 3.3 Enriched inputs to the model

Since the first works on the physics-informed framework, many of its aspects have been challenged and optimized to yield better results on various academic application cases. However, to the best of

our knowledge, there has been little to no discussion on optimizing the inputs of such models. The enriched physical knowledge of the problems have been applied to the models' architecture and inside the loss function to optimize the training, but no discussion have been conducted on using this knowledge to enrich the models' input fields. This limitation of the framework is significant in terms of both global performance and generalization capabilities. Providing information on changes, to the source term or boundary conditions for example, as input to the model would be beneficial, enabling it to expand its range of application without requiring re-training for every modification of the initial problem. This limitation is strongly linked with the failure of automatic differentiation pointed out in Theorem 3.2, since any additional input to the model leads to wrong evaluations of spatial derivatives, therefore to wrong estimations of PDE residuals.

Theorem 3.2 provides a formal proof of the failure of automatic differentiation as a tool to compute spatial derivatives on the fairly general setting of Assumption 3.1.

**Assumption 3.1** *Let $\Omega$ be an open, connex and smooth domain of $\mathbb{R}^d$ ($d \geq 1$), in which one needs to solve a PDE. Suppose that there exists a smooth scalar field $\varphi$ on $\Omega$ which is not constant in each direction: $\forall x = (x_1, \ldots, x_d)^T \in \Omega, \forall 1 \leq i \leq d, \frac{\partial \varphi}{\partial x_i} \neq 0$. Suppose a physics-informed model $M : \mathbb{R}^{d+1} \to \mathbb{R}$ takes as input the coordinates $x$ and the additional scalar field $\varphi$. Additionnaly, suppose that the computation of $\varphi$ as a function of $x$ has not been done inside the deep learning framework. Therefore, the computational graph of the automatic differentiation process will not record any dependency between $\varphi$ and $x$.*

Regarding the last part of the assumption, this statement is mostly verified in general, industrial settings, where complex geometries prevent the construction of analytical fields. It could also be verified when some additional information to the solution is provided in the form of data points, obtained from observations. For instance, as described in Section 3.1, when dealing with a Solid Mechanics problem, physical data such as material properties may be available only in a tabular form inside the domain.

**Theorem 3.2** *With the hypotheses made in Assumption 3.1, the partial derivatives of $M$ with respect to the coordinate $x$ as computed by automatic differentiation will not correspond to the physical spatial derivatives of the scalar field $M(x, \varphi(x))$.*

**Proof 3.3** *The proof is given in Appendix A.*

The recent development of neural operators (Li et al., 2020a; Lu et al., 2019; Li et al., 2020b) offers promising perspectives in the domain of scientific machine learning, since this class of models offers better generalization capabilities compared to plain neural networks. The aim of these models is to learn an operator mapping a PDE parameter (initial or boundary conditions, PDE coefficient or other) to the corresponding solution, rather than learning a single solution of a given PDE. The generalization capacities of these models are improved, since they do not require to be re-trained for each PDE instance that must be solved. However, few works directly use this class of operators within a physics-informed framework. One of the reasons of the difficulty to implement physics-informed neural operators is a direct consequence of Theorem 3.2: the PDE parameter given as input to the model should be constructed analytically in order to compute the PDE residuals with AD, therefore preventing the use of these models to real-life problems. For instance, Wang et al. (2021b) presented results based on analytic data, and Li et al. (2024) proposed function-wise differentiation as an alternative to AD. Such works could be adapted within the AD-based Physics-Informed framework, for instance by fitting tabulated data with splines or other interpolation techniques, however this would involve an additional layer of preprocessing complexity.

## 4 SPATIAL GRADIENT COMPUTATION AS A SPARSE NUMERICAL OPERATOR

As an alternative to AD, in the proposed hybrid numerical physics-informed model, the spatial derivative computations are handled by a numerical solver, which can be very general (finite difference, finite element, ...). These numerical operations are algebraic computations consisting of mainly matrix-vectors multiplications. Therefore, once the operator is built from the discretized domain, in can be directly integrated in the machine learning framework, both for the forward and

Domain & PDE $\xrightarrow{x,\ \varphi(x)}$ Model $M_\theta$ $\xrightarrow{M_\theta(x,\varphi(x))}$ Numerical diff. $\xrightarrow[\frac{\partial \nabla_x M_\theta}{\partial M_\theta}]{\nabla_x M_\theta}$ Automatic diff. $\longrightarrow$ $\frac{\partial \mathcal{L}}{\partial \theta}$

Figure 1: Main steps of the proposed hybrid approach of spatial gradient computations. The purple squares refer to computations made inside the deep learning framework and recorded by the automatic differentiation graph, while the blue square refers to the outside computation of derivative, with any numerical gradient kernel (finite difference, finite element,... ).

backward calls. Moreover, this sparse matrix-vector multiplication is independent of the model's complexity, which makes it far less expensive in terms of computational cost. On the other hand, the loss calculation does not require a call to AD, which drastically simplifies the computational graph. This further reduces the training time, which we demonstrate in Section 6. A step-by-step decomposition of the inclusion of the proposed framework in the Physics-Informed training is given in Appendix B. Figure 1 illustrates the computation of the loss derivative with the proposed hybrid numerical approach.

Our main experiment is presented in Section 6, and it has been realized with a hybrid Finite Element-Physics Informed approach, therefore we provide a detailed explaination of the gradients computation with this framework.

On a domain $\Omega \subset \mathbb{R}^n$ where a mesh has been built, we consider a nodal field $u = (u_1, \ldots, u_N)$, $N$ being the number of nodes. For the node $x_i$, $1 \le i \le N$, $u_i$ is the value of the field $u$ on the node $x_i$. The $P1$ finite element approximation is based on the consideration of the set of piecewise linear functions $(\varphi_i)_{1 \le i \le N}$, such that $\varphi_i(x_j)$ is equal to 1 if $i = j$, and 0 otherwise. The nodal field $u$ is therefore approximated by $u \approx \sum_i u_i \varphi_i$.

The gradient approximation of $u$ is computed as:

$$\nabla u \approx \mathcal{G}u, \quad \mathcal{G}_{i,j} = \frac{1}{\int_{\Omega_i} d\Omega} \int_{\Omega_i} \sum_{g=1}^{n_g} \nabla \varphi_j(x_g) d\Omega \tag{11}$$

Here, $\Omega_i$ is the set of elements surrounding node $i$, $x_g$ the Gauss points associated to these elements and $n_g$ the number of elements. This formula recovers 2 steps: the actual $P0$ gradient computation, and the gradient smoothing to recover a $P1$ field. Numerically, applying the operator $\mathcal{G}$ corresponds to a sparse matrix-vector multiplication in terms of computational complexity.

## 5 VALIDATION OF THE HYBRID NUMERICAL APPROACH

As proposed in Section 4, in the current framework the gradient computations could be handled by outside numerical operators. This flexibility allows for the use of complex operators with high accuracy, without having to implement them inside the deep learning framework. However, for simple cases, the use of in-place operators such as finite difference operators could still be efficient. In that case, there is no need to extract the numerical operator and to convert is as a sparse matrix, since in-place operators are traced back inside the computational graph. In the following section, a simple one-dimensional geometry will be used, as a proof of concept of the approach. The deep learning framework used is Pytorch (Paszke et al., 2019).

### 5.1 A NUMERICAL EXAMPLE OF A FAILURE OF AUTOMATIC DIFFERENTIATION

To illustrate the limitation of AD pointed out in Section 3.3, the setting of Assumption 3.1 has been reproduced. A simple neural network has been initialized with random weights, and two inputs are given to the model: a position field $x$, of evenly placed points in $(0, 1)$, and another input field $\varphi$, given by $\varphi(x) = \sin(10\pi x)$. The computation of $\varphi$ has been done outside of the AD computational graph. The output of the neural network, $M(x, \varphi(x))$, is then differentiated, both by AD and by a finite difference method. To further validate the finite difference computation, two operators have been used: one inside the Pytorch framework, with the `torch.gradient`

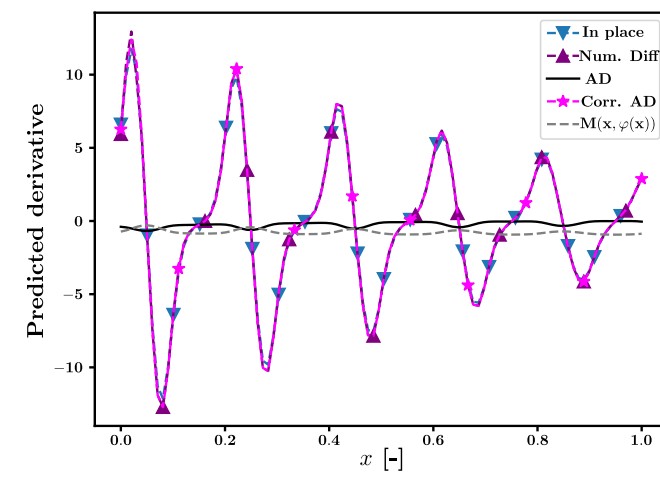

Figure 2: Derivative of the neural network's output with respect to $x$. 'In place' and 'Num. Diff' refer to the two finite difference derivative estimations, respectively computed by `torch.gradient` and `findiff`. 'AD' is the automatic differentiation computation, and 'Corr. AD' is the corrected automatic differentiation computation, to account for the missing terms of equation 20.

function, and one with another package, `findiff` (Baer, 2018), to illustrate the call to outside numerical operators. Moreover, the error made by AD when computing physical derivatives has been identified, and corresponds to the gap between the quantities written in (20) and (21). Therefore, in this simple case, it is straightforward to correct the derivative computed by AD to account for the missing term. This corrected derivative has been computed as well. The result is given in Figure 2.

As expected, the derivative computed by AD does not correspond to the actual derivative of $M(x, \varphi(x))$. On the other hand, the two numerical methods approximations are close, and are also close to the corrected analytical derivative. This result illustrates the limitation of AD on a simple setting. While a correction of the AD prediction is possible in this case, this should not be true in general, when analytical derivatives of $\varphi$ cannot be computed.

### 5.2 A ONE-DIMENSIONAL DIFFUSION EQUATION

In this section, a simple one-dimensional diffusion equation is considered, to illustrate the discussions of Sections 3.1 and 3.2. The considered domain is $\Omega = (0, 1)$, and the PDE is the following:

$$-(\alpha(x)u'(x))' = 0, \quad x \in \Omega, \quad \alpha(x) = 0.1 + x, \tag{12}$$
$$u(0) = 1, \quad u(1) = -1. \tag{13}$$

The solution to this problem is $u^*(x) = A \log_{10}(0.1 + x) + (1 + A)$, where $A = {}^{-2}/\log_{10}(11)$.

Two experiments have been conducted for this problem. In the first one, the diffusion coefficient $\alpha$ is computed and tabulated. In the second one, the Dirichlet boundary conditions are strongly imposed, as described in Section 3.2. In both cases, a plain PINN has been trained, and the results are compared with a model with the same architecture, but trained with our hybrid approach. The model is a multi-layer perceptron (MLP) with 3 hidden layers of width 50. The tanh activation function has been used. The trainings have been conducted with the Adam optimizer for 25,000 epochs, on a single CPU Intel Core i7 with 32Go of RAM, with a learning rate of $5 \times 10^{-3}$. The results are displayed in Figure 3.

While our proposed model converges to the true solution in both cases, the plain PINN does not. In the case of a tabulated coefficient $\alpha$, the model converges to a linear function, which would be solution to equations (12) - (13) if the coefficient $\alpha$ was constant. For the strong imposition of

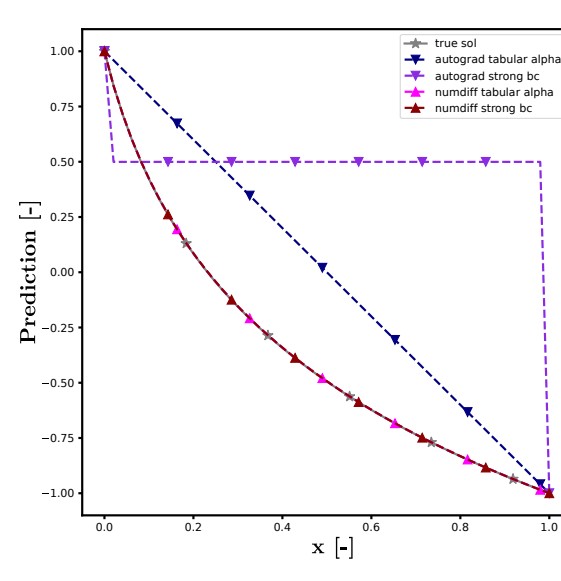

Figure 3: Predicted solution to (12) - (13). Grey: analytical solution. Predicted solution of the plain PINN, for tabulated $\alpha$ (Blue) and strong imposition of Dirichlet Boundary conditions (Purple). Predicted solution of our hybrid numerical PINN, for tabulated $\alpha$ (Pink) and strong imposition of Dirichlet Boundary conditions (Dark Red).

Dirichlet boundary conditions, the plain PINN converges to a constant function, which would be solution to equations (12) - (13) if the boundary conditions were not imposed.

# 6  A TWO-DIMENSIONAL STATIC LINEAR ELASTICITY PROBLEM

## 6.1  PROBLEM PRESENTATION

To generalize our approach to a more complex setting, we have addressed a two-dimensional static problem. The target field is the vector displacement, with Dirichlet boundary conditions on the domain $\Omega$ plotted in Figure 4 and representing the Olympic rings. The target function $\mathbf{u}^* = (u_x^*, u_y^*)$ is obtained with Finite Element Method (FEM). The mathematical formulation of the problem is the following:

$$\operatorname{div} \boldsymbol{\sigma}(\boldsymbol{\varepsilon}) = \mathbf{0}, \tag{14}$$

$$\boldsymbol{\varepsilon}(\mathbf{u}) = \nabla \mathbf{u} + \nabla \mathbf{u}^T, \qquad \boldsymbol{\sigma}(\boldsymbol{\varepsilon}) = \lambda \operatorname{Tr}(\boldsymbol{\varepsilon})\mathbf{I} + 2\mu\boldsymbol{\varepsilon}, \tag{15}$$

$$\mathbf{u}(x, y) = \mathbf{u}^*(x, y), \qquad (x, y) \in \Gamma \subset \Omega. \tag{16}$$

The Lamé parameters $\lambda$ and $\mu$ are set to 1, and $\mathbf{I}$ denotes the identity matrix. A mesh was built on this complex geometry and is made of 5104 nodes. The FEM computation took 0.21 second and 1452 iterations for a relative tolerance of $10^{-4}$, and 0.4 second (2843 iterations) for a relative tolerance of $10^{-8}$ using a preconditioned conjugate gradient method with Jacobi preconditioner. The experiments were made on a single Intel Xeon Gold CPU. The FEM gradient kernel has been used for our hybrid approach, following the framework presented in Section 4.

## 6.2  RESULTS AND DISCUSSION

Three models were trained on this problem: one with our hybrid approach with strong imposition of boundary conditions, and two models trained with AD: one with strongly enforced boundary

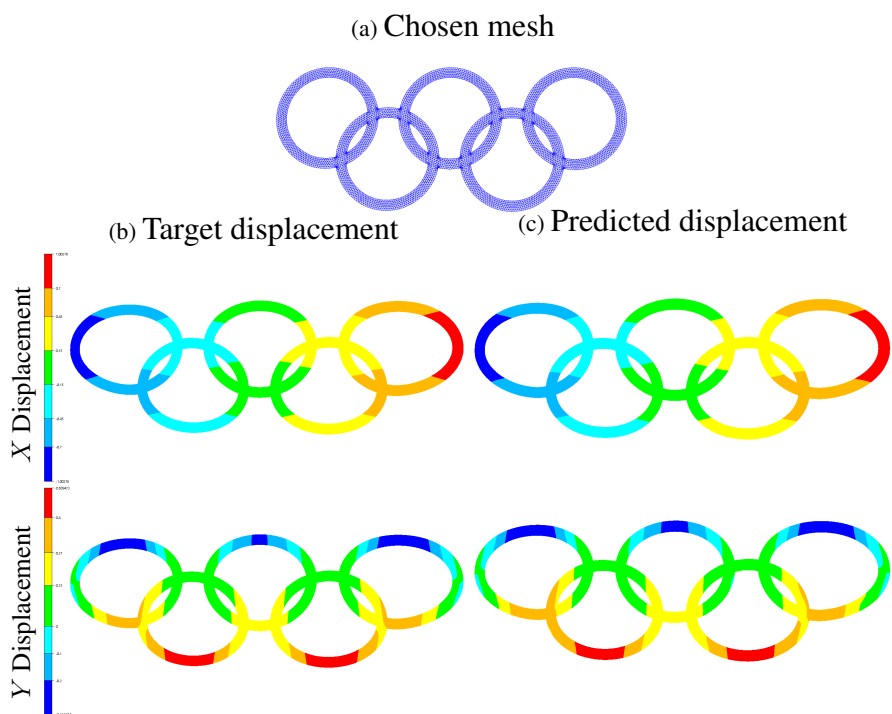

(a) Chosen mesh

(b) Target displacement    (c) Predicted displacement

Figure 4: Meshed domain $\Omega$, prediction of our model and target displacement. The boundary nodes where the target $X$ displacement is equal to -1 and 1 are the nodes with Dirichlet boundary conditions.

conditions following our approach, and one with a weak imposition of this constraint. The training has been conducted for 20000 epochs with the Adam optimizer, and a learning rate of 0.005. The models are Multi-Layer Perceptrons with 3 hidden layers of width 100, and with the Tanh activation function. The results and the training time are presented in Table 1.

Table 1: Results on the linear elasticity case. The relative Mean Squared Error is reported. 'Hybrid FE-PINN' refers to our hybrid Finite Element (FE) PINN. 'AD PINN, Strong BC' (resp.'AD PINN, Weak BC') refers to the AD PINN, with strongly (resp. weakly) enforced boundary conditions.

| Model | Relative error (%) | Training time (s) |
|---|---|---|
| Hybrid FE-PINN (ours) | **0.05** | $4.97 \times 10^2$ |
| AD PINN, Strong BC | 19300 | $1.80 \times 10^3$ |
| AD PINN, Weak BC | 94 | $1.82 \times 10^3$ |

With a relative mean squared error of 0.05%, our model can accurately reconstruct the target solution; the predicted solution is plotted in Figure 4. In contrast, the AD model trained with weak imposition of boundary constraints only achieves a relative error of 94% due to the complexity of the geometry. The AD method with strongly enforced boundary conditions performs even worse, with a relative error of 19300%, demonstrating the difficulty of strongly enforcing boundary conditions with plain PINNs. While more complex models like Fourier Neural Operators or Graph models could have better accuracy than plain neural networks, this case highlights the performance of our hybrid approach, and its ability to address real-life geometries and equations.

To further demonstrate the competitiveness of our method in terms of computational complexity, we performed forward and inverse gradient computations using AD and FEM on the same geometry. Fifty runs were conducted for each computation, and the standard deviation is reported. Two models were used: a shallow neural network (one hidden layer of width 64) and a deep one (twenty layers

of width 256). The average computation time and the associated standard deviation are reported in Figure 5.

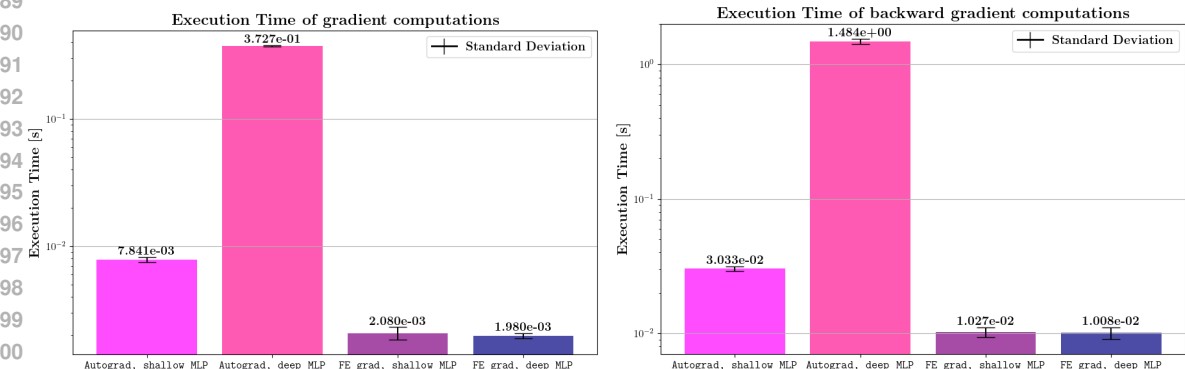

Figure 5: Execution time of forward (left) and backward (right) gradient computation. The four values correspond to the computation with AD, for the shallow (magenta) and deep (pink) models, and with our FE gradient computation, for both models (purple and navy).

Our method is slightly faster than AD on the shallow network, but the main improvement is seen with the deep network. As expected, our method is independent of the model's complexity, whereas the AD technique needs to trace back all intermediate operations to compute the final physical gradient. Therefore, for more complex models, our numerical differentiation technique significantly outperforms AD: there is a speed-up of up to almost 180 and 150 times for the forward and backward pass respectively. The computational graph of our model is also way simpler, since no backtracking is needed for the loss computation. This simplification, combined with the results presented in Figure 5, explain the shorter training time of our hybrid method compared to AD PINNs.

## 7 CONCLUSION AND PERSPECTIVES

In this paper, a novel framework, hybrid numerical PINNs, has been presented. This general procedure is flexible and can be combined with any improvement regarding the model's architecture or the training step. Our hybrid approach allows to strongly impose Dirichlet boundary conditions on arbitrary shapes, with no preprocessing complexity. This new framework overcomes the limitations of automatic differentiation, paving the way for further enhancements to physics-informed applications. The new capabilities have been demonstrated on several numerical problems.

Important future directions include the extension of this setting to more complex problems, for instance with more challenging boundary conditions and equations, and its application with state-of-the-art models such as Fourier Neural Operators or Graph Networks.

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

## A  PROOF OF THEOREM 3.2

**Proof A.1 (Proof of Theorem 3.2)** *The proof is a straightforward application of the chain rule, which is detailed here for completeness. First, define*

$$\psi \colon \mathbb{R}^d \longrightarrow \mathbb{R}^{d+1}$$
$$x \longmapsto (x, \varphi(x)).$$

*For $1 \leq i \leq d$ and $x \in \mathbb{R}^d$, the partial derivative of $M(x, \varphi)$ with respect to $x_i$ is:*

$$\frac{\partial M(x, \varphi(x))}{\partial x_i} = \frac{\partial (M \circ \psi)}{\partial x_i}(x) \tag{17}$$

$$= \sum_{j=1}^{d+1} \partial_j M \times \frac{\partial \psi_j}{\partial x_i}, \tag{18}$$

*where $\partial_j M$ is the partial derivative of $M$ with respect to its $j$-th input component. Moreover, by definition of $\psi$, for $1 \leq j \leq d+1$:*

$$\frac{\partial \psi_j}{\partial x_i} = \begin{cases} 1 & \text{if } j = i, \\ \frac{\partial \varphi}{\partial x_i} & \text{if } j = d+1, \\ 0 & \text{otherwise.} \end{cases} \tag{19}$$

*Combining 17 and 19, we finally get:*

$$\frac{\partial M(x, \varphi(x))}{\partial x_i} = \partial_i M + \partial_{d+1} M \times \frac{\partial \varphi}{\partial x_i}. \tag{20}$$

*On the other hand, since, by Assumption 3.1, the computational graph of the automatic differentiation process does not record any dependency between $\varphi$ and $x$, the derivative of $\varphi$ with respect to $x_i$ will be recorded as null. Therefore, the automatic differentiation computation will simply yield the result:*

$$\frac{\partial M(x, \varphi(x))}{\partial x_i}\bigg|_{\text{auto. diff.}} = \partial_i M. \tag{21}$$

*The difference between equation 20 and equation 21 concludes the proof.*

## B  STEP-BY-STEP DECOMPOSITION OF THE PROPOSED FRAMEWORK

We consider a model $M_\theta$, such as a neural network, $\theta$ being its trainable parameters. $M_\theta$ takes as input the position coordinates $x$, along with any additional field $\varphi(x)$. We focus on a generic first order static PDE, and we detail the main steps of the loss residuals computation. Note that this discussion can be generalized to any time-dependent PDE. The considered PDE on any domain $\Omega$ is the following:

$$\mathcal{N}(u, \nabla u) = 0. \tag{22}$$

We focus on the PDE residuals loss, therefore the boundary conditions are omited. Given $N$ sample points $x_1, \ldots, x_N$ inside the domain $\Omega$, the PDE residuals loss is computed as follows:

$$\mathcal{L}(\theta) = \frac{1}{N} \sum_{i=1}^{N} \|\mathcal{N}\left[M_\theta(x_i, \varphi(x_i)), \nabla M_\theta(x_i, \varphi(x_i))\right]\|^2 := \frac{1}{N} \sum_{i=1}^{N} \mathcal{L}_i[M_\theta, \nabla M_\theta]. \tag{23}$$

Here, $\mathcal{L}_i$ is the loss term corresponding to the collocation point $x_i$. In order to perform gradient-based training, the gradient of the loss with respect to the model's parameters $\theta$, $\frac{\partial \mathcal{L}}{\partial \theta}$, needs to be computed. The computation is made as follows:

$$\frac{\partial \mathcal{L}}{\partial \theta} = \frac{1}{N} \sum_{i=1}^{N} \left( \frac{\partial \mathcal{L}_i}{\partial M_\theta} \times \frac{\partial M_\theta}{\partial \theta} + \frac{\partial \mathcal{L}_i}{\partial \nabla M_\theta} \times \frac{\partial \nabla M_\theta}{\partial M_\theta} \times \frac{\partial M_\theta}{\partial \theta} \right). \tag{24}$$

Each term of equation 24 is computed as follows:

- $\frac{\partial \mathcal{L}_i}{\partial M_\theta}$ and $\frac{\partial \mathcal{L}_i}{\partial \nabla M_\theta}$: Both these terms directly depend on the loss considered in equation 23, and the PDE (22). Once provided with an approximation of $\nabla M_\theta$, the loss computation is straightforward, and these derivatives are computed by automatic differentiation.

- $\frac{\partial \nabla M_\theta}{\partial M_\theta}$: This term corresponds to the derivative of the numerical gradient computation. Once this operator is stored as a sparse matrix, its backward call can be computed by AD.

- $\frac{\partial M_\theta}{\partial \theta}$: This term is simply the derivative of the model's output with respect to its parameters, computed by AD.

