# OpenReview forum: "Hybrid Numerical PINNs: On the effectiveness of numerical differentiation for non-analytic problems"
_ICLR.cc/2025/Conference — ICLR 2025 Conference Withdrawn Submission_

### Official Review · Reviewer_tLLa · 2024-10-28

**Soundness:** 3
**Presentation:** 2
**Contribution:** 3
**Rating:** 6
**Confidence:** 2

**Summary:**

This paper addresses the limitations of automatic differentiation (AD) in PINNs for non-analytic PDEs. The authors propose a hybrid approach that combines numerical solvers with deep learning models to replace AD for gradient calculations. The proposed method enables the exact imposition of Dirichlet boundary conditions. The proposed approach is flexible and can be incorporated into any physics-informed model. Gradient computation is up to two orders of magnitude faster than automatic differentiation.

**Strengths:**

- **Addresses AD Limitations**. The proposed method addresses cases where AD fails, e.g., when the PDE coefficients do not have an analytic form, or when enriched input data is fed to the deep learning model.
- **BC Imposition**. The proposed method allows strong constraints of Dirichlet BCs.
- **Scalability and Efficiency**. The proposed approach shows significant speedups in gradient computation compared to AD-based PINNs. The numerical cost is independent of the complexity of the trained model.
- **Flexiblility**. The proposed approach is flexible and can be incorporated into any physics-informed model.

**Weaknesses:**

- **Generalization**. The proposed work could handle 1D and 2D PDEs with Dirichlet BCs. Yet, whether it could generalize to higher dimensions or more complex BCs remains unknown.
- **Dependency on External Numerical Solvers**. The reliance on external numerical solvers makes the model more complex.
- **Insufficient PINN Baselines**: The experiments do not thoroughly compare with SOTA neural operators (e.g., FNO, GNO), which are considered an important baseline for PINN-based neural routines.
- **Presentation**. The presentation of this paper could be further improved. There are also typos (e.g., Eq. (19) is not properly aligned).

**Questions:**

- How does the proposed method perform on 3D PDEs or larger-scale problems?
- Can the method handle more complex boundary types (e.g., Neumann, Robin) just as effectively?
- Handling of Nonlinear Operators: How well does the hybrid method generalize to nonlinear PDEs?

---

> ### Author Response · Authors · 2024-11-18
> **Answer to reviewer tLLa**
>
> First, we would like to thank you for your review, and for your constructive comments. Here is our answer to your questions and concerns.
>
> How does the proposed method perform on 3D PDEs or larger-scale problems?
>
> Handling of Nonlinear Operators: How well does the hybrid method generalize to nonlinear PDEs?
>
> Although further experiments are needed for more complex cases, the initial results on complex geometries are promising regarding our model’s ability to address such problems. Moreover, FEM has proven to be efficient in many highly non-linear problems. The range of problems they can solve is, for now, broader than the PINNs capacities, as detailed in [1]. Their use in a hybrid approach such as the one we introduce should therefore not restrict the ability of our model to deal with complex, non-linear PDEs.
>
>
> [1] Can physics-informed neural networks beat the finite element method? Grossmann et al., Journal of Applied Mathematics, 2024.
>
>
> Can the method handle more complex boundary types (e.g., Neumann, Robin) just as effectively?
>
>
> Even though strong imposition of more complex boundary types may not be as straightforward, our method offers the possibility to handle them in the same fashion as for traditional Physics-Informed models. For instance, one could compute the error between the predicted derivative on the boundary and the target Neumann or Robin condition, and include it inside the loss, as an additional term. Alternatively, variational approaches could be used, since these conditions integrate well into such formulations. This has already been implemented in several Physics-Informed models, and it could directly be adapted to our case, without losing the speed-up gained with our gradient computation.
>
>
> More generally, we would like to emphasize that our loss computation can be combined with any model or problem formulation, including operator learning, with the use of FNO or GNO. In this work, we have restricted our models to simple neural networks to emphasize the benefits of  our approach, rather than aiming at achieving state of the art results on academic benchmarks.

---

> ### Comment · Reviewer_tLLa · 2024-11-26
>
> Thank you for your clarification. I am keeping my original rating.

---

### Official Review · Reviewer_SsPY · 2024-10-31

**Soundness:** 1
**Presentation:** 3
**Contribution:** 1
**Rating:** 3
**Confidence:** 4

**Summary:**

The authors highlight certain limitations of automatic differentiation in computing PDE residuals used in physics-informed training. The solution is to compute residuals with classical discretization instead and use automatic differentiation only to compute derivatives with respect to neural network parameters.

**Strengths:**

The article is easy to read. The motivation of the authors and the experiments conducted are clearly explained. I also would like to thank the authors for reporting the time needed to obtain a solution with PiNN and by preconditioned conjugate gradient. This level of honesty feels refreshing.

**Weaknesses:**

There are two problems:
1. Major literature omissions
2. Examples showing fail of automatic differentiation are contrived

I will elaborate on them in the next section.

**Questions:**

I think it, is more instructive to focus on major problems. Typos and minor issues with the presentation will be completely ignored.

1. **Major literature omissions.**

   Many works have been available for quite some time that pursue the same research direction as the one chosen by the authors. At the very least authors should mention these works, better, an extended discussion, supported by numerical results, should be added that explicitly compares the method proposed in this work with the related ones. Below one can find several examples of articles that are related to the content of a current contribution.

   a. Finite element method-enhanced neural network for forward and inverse problems, https://amses-journal.springeropen.com/articles/10.1186/s40323-023-00243-1. Here finite element discretization is combined with a neural network operating over the discretized grid.

   b. Physics Informed Neural Network using Finite Difference Method, https://ieeexplore.ieee.org/abstract/document/9945171. In this paper, the authors use finite difference approximation with PiNNs.

   c. Hybrid Finite Difference with the Physics-informed Neural Network for solving PDE in complex geometries. https://arxiv.org/abs/2202.07926. Again, in this paper finite difference method is applied to enforce residual.

   d. hp-VPINNs: Variational Physics-Informed Neural Networks With Domain Decomposition. https://arxiv.org/abs/2003.05385 - the well-known article where authors use neural networks for trial space and polynomials for test space. This is again an example of classical discretization applied in conjunction with PiNNs.

   I suggest authors comment on the articles above and their relation to the content of a current contribution. Besides that, I would recommend performing a more thorough literature review to include other relevant contributions.

2. **Examples showing failure of automatic differentiation are contrived.**

   The authors demonstrate three examples of when automatic differentiation fails. I will argue that all of them are highly artificial.

   **Tabulated data.**

   In this example diffusion coefficient $\alpha(x)$ appears in the conservative formulation of ODE $-\frac{d}{dx} \left(\alpha(x) \frac{d u}{dx}\right) = 0$ is known only in selected set of points. If one naively computes residual by automatic differentiation in these points, the resulting PiNN loss leads to a completely wrong solution. I have several objections:
   1. Under this scenario classical discretization method is also not defined unless one adapts the grid to the special locations where $\alpha(x)$ is known. Does it mean this example is problematic for classical methods too?
   2. A usual solution for this situation is known as interpolation and in more complex situations as data assimilation (a good example is https://www.ecmwf.int/en/forecasts/dataset/ecmwf-reanalysis-v5). This problem is well-studied and can be solved by various means. Can the authors try, say, cubic splines and report the result of PiNN training? To do that one defines $\alpha(x)$ as a function that performs cubic spline interpolation given its values at the known points, after that automatic differentiation can be used to compute derivatives. For that to work interpolation should be differentiable. Alternatively, derivatives can be estimated and tabulated. After that one can rewrite the equation in non-conservative form and run automatic differentiation for other parts of the residual.

   **Strong imposition of Dirichlet boundary conditions.**

   In this example, authors construct a mask that defines a physical domain and use it to exactly impose Dirichlet boundary conditions. There is a natural way to enforce boundary conditions for PiNNs exactly. The techniques are explained in "Exact imposition of boundary conditions with distance functions in physics-informed deep neural networks", https://arxiv.org/abs/2104.08426. One way to do that is to use Floater's mean value coordinates to form a smooth distance function for a given boundary and use, say, RBF (or transfinite interpolation, linear interpolation, or ordinary least squares, etc) to enforce the desired value on the boundary. The final ansatz reads $NN(x)\phi(x) + g(x)$, where $\phi(x)$ is smooth and vanishing on the boundary and $g(x)$ reproduces boundary conditions exactly.

   **Enriched input to the model.**

   In this hypothetical scenario, the authors suggest that it would be hard to provide the model with additional inputs and keep using automatic differentiation. The authors correctly mentioned that this is done in the domain of operator learning. The reason why automatic differentiation is hard is as follows "However, few works directly use this class of operators within a physics-informed framework. One of the reasons for the difficulty of implementing physics-informed neural operators is a direct consequence of Theorem 3.2: the PDE parameter given as input to the model should be constructed analytically to compute the PDE residuals with AD, therefore preventing the use of these models to real-life problems. For instance, Wang et al. (2021b) presented results based on analytic data, and Li et al. (2024) proposed function-wise differentiation as an alternative to AD." I find this presentation problematic:
   1. DeepONet can be trained on tabular data either with interpolation or when the formulation is not conservative, e.g., in the cited DeepONet paper this is the case for all PDEs considered: Burgers, diffusion-reaction, advection, eikonal.
   2. The reason FNO is not using automatic differentiation is that its process functions are explicitly discretized on a uniform grid. In other words, FNO is a function over functions. It is not possible to apply automatic differentiation because these functions do not have computation graphs in the usual sense.

To summarise, in my opinion, all given examples of automatic differentiation failure are highly unnatural. Any practitioner with basic knowledge of automatic differentiation will never try to arrange computations as the authors suggest. It seems to me that these modes of failure are too obvious and easily avoidable.

I am ready to change my opinion, if authors come up with convincing arguments, so I encourage authors to address my concerns in the rebuttal.

---

> ### Author Response · Authors · 2024-11-18
> **Answer to Reviewer SsPY**
>
> First, we would like to thank you for your very thorough review, and for the time you spent on our work. Here is our answer to your remarks and questions.
>
> $1.$ Litterature omissions
>
> We would like to thank you for the relevant articles you cited, they have been added to our review, along with several others, and we also added a comparison with our approach. We acknowledge that some similar works have been published in recent years, some of them with very convincing results. For the hybrid Finite Elements – Physics Informed works, to the best of our knowledge, every published paper, including the one you cited, used the full Finite Element formulation, which necessitates the computation of the mass matrix for the PDE variational formulation. We argue that our approach, in which only the gradient operator is computed, is more flexible, since this operator can be used for any kind of differential operator, for a fixed domain.
>
> In our work, we emphasize that such hybrid methods offer significative speed-ups in gradient computations compared to automatic differentiation, regardless of the model’s complexity. We also show that many recent improvements on PINNs (strong imposition of boundary conditions and balancing loss terms for instance) can be integrated seamlessly in such hybrid frameworks. We argue that these improvements are not restricted to our approach, but that they are simplified drastically.
>
> $2.$ Regarding the examples we pointed out on the failures of automatic differentiation, we would like to clarify that the given examples are indeed straightforward problems, that could be easily solved with the current autodiff-based PINN framework, with an additional step of preprocessing complexity. Rather than showing scenario that would be unreachable by AD PINNs, our point was to illustrate the previous theoretical discussions on the flexibility of the hybrid approach. We provide further details on the three cases.
>
> $2a.$ Tabulated data
>
> In numerical analysis, usually, once the domain is meshed, external material solvers compute quantities of interest such as $\alpha(x)$ on the mesh nodes, rather than just having tabulated data on random points. Such solvers may not be differentiable; therefore this data cannot be incorporated as such inside the PINN framework. Interpolating this data, with splines for instance, would indeed solve this issue, but it comes with an additional level of preprocessing complexity. Morevoer, such interpolation may be non-trivial for irregular domains, with highly heterogeneous materials or conditions.
>
> $2b.$ Dirichlet boundary conditions
>
> Many works have been published regarding the strong imposition of Dirichlet boundary conditions with PINNs, including the work you mention. However, once again, it is nontrivial to fit a distance function to impose the ansatz $NN(x) \varphi(x) + g(x)$, specially in complex, three-dimensional domains. The distance function must also be computed inside the AD framework to be differentiable, which restricts the possibility to call outside operators for such computations. Once again, the recent progress of PINNs shows that these difficulties can be overcome in most cases, however we think that the way we directly impose such conditions, with no additional preprocessing complexity, is of interest for the PINN community. To the best of our knowledge, such discussions have not been carried out before.
>
> $2c.$ Enriched inputs to the model
>
> We understand your concerns regarding our formulation, which could be misleading. We did not mean to say that the cited works could not be used in the traditional Physics-Informed formulation, but rather that such formulation comes with added complexity, as explained in the two previous points. We have changed this sentence in the updated version of our paper, and we thank you for your constructive feedback.
>
>
> Finally, we would like to emphasize that we see the main contribution of our work to be the speed-up offered by our numerical gradient computation, and the way many recent improvements of the Physics-Informed framework can be simplified with the use of numerical gradient kernels, whether they are built from Finite Element, Finite Difference or any other numerical tool. This is especially true for the strong imposition of Dirichlet boundary conditions, which has been the subject of many recent works on the Physics-Informed framework, and which can be solved effortlessly with the framework we propose. An additional benefit is the smoother and shorter training, provided by the simplification of the computation graph: indeed, no calls to automatic differentiation are needed for the forward computation of the loss.

---

> > ### Comment · Reviewer_SsPY · 2024-11-21
> >
> > ```
> > For the hybrid Finite Elements – Physics Informed works, to the best of our knowledge, every published paper, including the one you cited, used the full Finite Element formulation, which necessitates the computation of the mass matrix for the PDE variational formulation. We argue that our approach, in which only the gradient operator is computed, is more flexible, since this operator can be used for any kind of differential operator, for a fixed domain.
> > ```
> > I respectfully disagree. There are works that uses pure finite differences. Among the ones that I cited is https://arxiv.org/abs/2202.07926 (see equation 5).
> >
> > ```
> > In numerical analysis, usually, once the domain is meshed, external material solvers compute quantities of interest such as  on the mesh nodes, rather than just having tabulated data on random points. Such solvers may not be differentiable; therefore this data cannot be incorporated as such inside the PINN framework. Interpolating this data, with splines for instance, would indeed solve this issue, but it comes with an additional level of preprocessing complexity.
> > ```
> >
> > Many modern solvers actually are built to provide continuous solutions. Familiar examples are spectral solvers, FEM, and dense output for dopri45. Most of them are differentiable, with some caveats.
> >
> > ```
> > Morevoer, such interpolation may be non-trivial for irregular domains, with highly heterogeneous materials or conditions.
> > ```
> >
> > When the domain is highly irregular, PiNNs face the same problems as all other methods. PiNNs are meshes, which removes the need for mesh generation, but the domain still needs to be described, including potentially complex surfaces where boundary conditions may be imposed. It is an illusion that PiNN is better than other methods. For example, one can consider classical RBFs if mesh free property is really important.
> >
> > ```
> > Many works have been published regarding the strong imposition of Dirichlet boundary conditions with PINNs, including the work you mention. However, once again, it is nontrivial to fit a distance function to impose the ansatz $NN(x) \varphi(x) + g(x)$, specially in complex, three-dimensional domains. The distance function must also be computed inside the AD framework to be differentiable, which restricts the possibility to call outside operators for such computations. Once again, the recent progress of PINNs shows that these difficulties can be overcome in most cases, however we think that the way we directly impose such conditions, with no additional preprocessing complexity, is of interest for the PINN community. To the best of our knowledge, such discussions have not been carried out before.
> > ```
> >
> > Suppose I have a complex domain with Dirichlet boundary conditions that explicitly depend on coordinates. For example, I am trying to find the distribution of temperature in a warehouse (3D, complex geometry) with prescribed temperatures on walls. Can the authors describe how to find $T_{D}(x)$ that fulfills Dirichlet boundary conditions? This is necessary for the proposed method to work.
> >
> > ```
> > Finally, we would like to emphasize that we see the main contribution of our work to be the speed-up offered by our numerical gradient computation, and the way many recent improvements of the Physics-Informed framework can be simplified with the use of numerical gradient kernels, whether they are built from Finite Element, Finite Difference or any other numerical tool.
> > ```
> >
> > Again, this is the same motivation as in https://arxiv.org/abs/2202.07926.

---

> > > ### Author Response · Authors · 2024-11-22
> > > **Answer to Reviewer SsPY's comment**
> > >
> > > *I respectfully disagree. There are works that uses pure finite differences. Among the ones that I cited is https://arxiv.org/abs/2202.07926 (see equation 5).*
> > >
> > > **We apologize, our sentence was unclear. We meant to say that while some works, such as the one you point out, use pure finite differences as a way of computing derivatives, to the best of our knowledge, there are no similar works using Finite Elements. The hybrid works using Finite Elements, unlike Hybrid Finite Differences, use the full formulation and therefore need to build the whole mass matrix.**
> > >
> > > *Many modern solvers actually are built to provide continuous solutions. (...) Most of them are differentiable, with some caveats.*
> > >
> > > **Many modern solvers are indeed continuous, and even differentiable to some extent, but computing the derivative of their output with respect to their input, which would be necessary to include them in a Physics-Informed framework, is not straightforward as no analytical derivative is available. Numerical procedures, such as the ones we introduce, are therefore required instead of the traditional AD framework.**
> > >
> > > *Regarding your remark on meshless PINNs*
> > >
> > > **We fully agree with your view, and we consider that the meshless property of PINNs is only valid for simple geometries that can be described analytically. In our method, we rely on a mesh to perform FE gradient computations, and we try to generalize the Physics-Informed approach to complex settings where meshless PINNs struggle (see, for instance, Fig. 4 and Table 1). We argue that in these problems, the mesh dependency of our method is not a main drawback.**
> > >
> > > *Regarding your Temperature example in the warehouse*
> > >
> > > **Actually, our method does not necessitate to compute an analytical formula for $T_D(x)$. Since the PDE residuals are computed numerically, the value on the border does not need to be traced back by AD. Therefore, once our model has predicted a solution for the temperature on the whole domain, we would simply modify the value of the prediction on the wall directly (for instance, if we know that on certain points of the wall, the temperature is equal to a given value, we would modify the output of the model at these particular points with the specified value). The loss computation would account for this modification, therefore allowing to converge to the PDE solution during training.**
> > >
> > > *Regarding the difference with [https://arxiv.org/abs/2202.07926](https://arxiv.org/abs/2202.07926)*
> > >
> > > **The article you mention does present a hybrid approach, with Finite Difference derivative computations. While they do achieve significative accuracy improvements compared to AD PINNs, we argue that their Finite Difference scheme prevents them to address more complex, challengiing geometries, where typically, Finite Element methods are mostly used. Moreover, to impose Dirichlet boundary conditions, they use signed distance functions to the border, which could be difficult to implement on more complex problems. As discussed in the previous point, our method does not require such preprocessing complexity, and benefits from the accuracy and versatility of Finite Element schemes.**

---

> > > > ### Comment · Reviewer_SsPY · 2024-11-25
> > > >
> > > > ```
> > > > Actually, our method does not necessitate to compute an analytical formula for $T_D(x)$. Since the PDE residuals are computed numerically, the value on the border does not need to be traced back by AD. Therefore, once our model has predicted a solution for the temperature on the whole domain, we would simply modify the value of the prediction on the wall directly (for instance, if we know that on certain points of the wall, the temperature is equal to a given value, we would modify the output of the model at these particular points with the specified value). The loss computation would account for this modification, therefore allowing to converge to the PDE solution during training.
> > > > ```
> > > >
> > > > I believe I have a better understanding of this point after your comment. Equation (9) suggests that $u_D(x)$ is a “bulk” field, but I agree it can be defined only on the boundary.
> > > >
> > > > I thank the authors for the clarifications on various points raised by me and other reviewers. Unfortunately, even after the discussion, my opinion remains largely unchanged: the problems with automatic differentiation studied in the article are artificial, and related hybrid approaches are available elsewhere.

---

### Official Review · Reviewer_wU62 · 2024-11-02

**Soundness:** 2
**Presentation:** 1
**Contribution:** 1
**Rating:** 1
**Confidence:** 5

**Summary:**

In this manuscript, the authors propose using numerical differentiation, instead of auto-differentiation, to deal with the physical derivatives of neural networks. The authors claim that numerical differentiation is faster than auto-differentiation. The authors also claim that it can solve several problems in the existing PINN framework. The authors conduct several numerical experiments in some cases.

**Strengths:**

It's really good that the authors attempt to find another way of doing differentiation in PINN. As stated in the manuscript, most works in this area focus on the network architecture or the loss function, but few focus on improving the pipeline defined by PINN or Neural Operator. Sometimes, it's really important to jump out from the existing framework to find a better way to solve the problem.

**Weaknesses:**

While it is important to identify some fundamental problems in the existing framework, the reviewer suggests that the author should check if these problems can be perfectly solved by the existing method. If so, the authors should first learn these existing methods. In this manuscript, the authors claim that the existing PINN framework has two weaknesses: 1. Auto-differentiation can not deal with tabulated coefficients, and 2. Auto-differentiation can not deal with the network using a scalar field as input. However, both of them can be perfectly solved by existing methods. For the first problem, one can use a smooth enough function to fit the tabulated coefficients and use this function as the coefficient function. For the second problem, defining the JVP/VJP function of the scalar field can perfectly solve the problem, which is available in most AutoDiff frameworks. Thus, the authors should first learn these methods and compare the proposed method with these solutions, which is missed in the current manuscript.

**Questions:**

Suggestions are listed in weakness.

---

> ### Author Response · Authors · 2024-11-18
> **Answer to reviewer wU62**
>
> First, we would like to thank you for your review. Here is our response to the concerns you raised.
>
> $1.$	Regarding the inability of auto-differentiation to deal with tabulated coefficients, fitting this discrete data with a smooth enough function would indeed solve the problem. However, this comes with an added preprocessing complexity, and it may be difficult to implement on complex geometries and cases, for instance with high heterogeneity in the considered materials. This added computational cost may become prohibitive in such scenarii.
>
> $2.$	For the addition of scalar fields as input, we are not sure to understand your proposed solution to define the JVP/VJP function of the scalar field. This definition would only make sense if we know a priori the analytical dependency of this scalar field with respect to the position $x$, so that we could then derive its Jacobian matrix. In such case, we could include it directly into the auto-diff framework. Our hypothesis was that such dependency was not known, for instance if the considered scalar field has been obtained by a numerical solver, outside of the auto-diff framework. In that case, the analytical derivatives of such field are not known, and defining its gradient or Jacobian matrix would necessitate the help of numerical differentiation kernels, which is precisely what we propose in our work.
>
> $3.$	More generally, we would like to emphasize that in our opinion, the main improvement of our hybrid approach is its flexibility to include and impose additional information such as boundary conditions directly, without the need to fit interpolation functions or distance functions to the boundaries. Another advantage of this hybrid approach is its speed-up compared to automatic differentiation: our method is up to 180 times faster in the experiments we conducted, due to a simplified computational graph. We acknowledge that the first experiments we provided were simple, toy examples to highlight the previous theoretical discussions, and were in no way a full example that would be unreachable to classical PINNs. We believe that the problem presented in section 6, along with the comparison of execution time of gradient computations (Fig. 5), are more relevant examples.
>
> We would like to thank you again for your honest review, and we ask you to take into account our response for your final rating.

---

### Note · Authors · 2025-03-19

I have read and agree with the venue's withdrawal policy on behalf of myself and my co-authors.

---

### Meta-Review · Area_Chair_DoFz · 2024-12-22

**Metareview:**

The paper propose a rather strange idea that automatic differentiation has limitations in the computation of derivatives for PINNS.
As shown by the reviewers (see Additional Comments on Reviewer discussion) this is not so true.
For example, for tabulated data you can just replace it by piecewise-linear function and interpolate it.
Thus, I don't think this is a reasonable contribution.

**Additional Comments On Reviewer Discussion:**

The reviewer SsPY provided useful references and critique of the paper. The authors provided the answer, but unfortunately in this case the situation seems to be rather clear.

---

### Decision · Program_Chairs · 2025-01-22

Reject